# An Optimized Strategy Based on Conventional Ultrasound for Diagnosing Metabolic Dysfunction-Associated Steatotic Liver Disease

**DOI:** 10.3390/diagnostics13233503

**Published:** 2023-11-22

**Authors:** Xiongcai Feng, Junzhao Ye, Hong Deng, Xin Li, Lishu Xu, Shiting Feng, Zhi Dong, Bing Liao, Zhiyong Dong, Bihui Zhong

**Affiliations:** 1Department of Gastroenterology, The First Affiliated Hospital, Sun Yat-sen University, No. 58 Zhongshan II Road, Yuexiu District, Guangzhou 510000, China; fengxc3@mail2.sysu.edu.cn (X.F.); yejzh@mail2.sysu.edu.cn (J.Y.); 2Department of Infectious Diseases, The Third Affiliated Hospital, Sun Yat-sen University, No. 600 Tianhe Road, Tianhe District, Guangzhou 510080, China; denghong2@163.net; 3Department of Gastroenterology, The Tenth Affiliated Hospital of Southern Medical University (Dongguan People’s Hospital), No. 78 Wandao Road, Wanjiang District, Dongguan 523059, China; 18924354583@163.com; 4Department of Geriatrics, Guangdong Provincial People’s Hospital, No. 106 Zhongshan II Road, Yuexiu District, Guangzhou 510000, China; xulishu70@163.com; 5Department of Radiology, The First Affiliated Hospital, Sun Yat-sen University, No. 58 Zhongshan II Road, Yuexiu District, Guangzhou 510000, China; fengsht@mail.sysu.edu.cn (S.F.); dongzh7@mail.sysu.edu.cn (Z.D.); 6Department of Pathology, The First Affiliated Hospital, Sun Yat-sen University, No. 58 Zhongshan II Road, Yuexiu District, Guangzhou 510000, China; liaob@mail.sysu.edu.cn; 7Department of Bariatric Surgery, The First Affiliated Hospital, Jinan University, No. 613 West Huangpu Avenue, Tianhe District, Guangzhou 510080, China

**Keywords:** diagnosis, metabolic associated steatotic liver disease, ultrasound

## Abstract

The inherent drawbacks of the conventional B-mode ultrasound for metabolic dysfunction-associated steatotic liver disease (MASLD) are poorly understood. We aimed to investigate the impact factors and optimize the screening performance of ultrasound in MASLD. In a prospective pilot cohort recruited from July 2020 to January 2022, subjects who had undergone magnetic resonance imaging-based proton density fat fraction (MRI-PDFF), ultrasound, and laboratory test-based assessments were included in the deprivation cohort. A validation cohort including 426 patients with liver histologic assessments from five medical centers in South China was also recruited. A total of 1489 Chinese subjects were enrolled in the deprivation cohort, and ultrasound misdiagnosed 62.2% of the non-MASLD patients and failed to detect 6.1% of the MASLD patients. The number of metabolic dysfunction components and the alanine aminotransferase (ALT) level were associated with a missed diagnosis by ultrasound (OR = 0.67, 95% CI 0.55–0.82 *p* < 0.001; OR = 0.50, 95% CI 0.31–0.79, *p* = 0.003, respectively). Compared with ultrasound alone, the new strategy based on ultrasound, in combination with measurements of the number of metabolic dysfunction components and ALT and uric acid levels, significantly improved the AUROC both in the research cohort and the validation cohort (0.66 vs. 0.84, 0.83 vs. 0.92, respectively). The number of metabolic dysfunction components and ALT and uric acid levels improved the screening efficacy of ultrasound for MASLD.

## 1. Introduction

Metabolic dysfunction-associated steatotic liver disease (MASLD) was recently introduced by multiple international societies to modify the potentially stigmatizing “nonalcoholic” and “fatty” in the old term of nonalcoholic fatty liver disease (NAFLD) and metabolic dysfunction-associated fatty liver disease (MAFLD) [1]. The epidemic of MAFLD has conferred a huge clinical burden on about one third of the global population [2]. As well as simple steatosis with histological signs of fat accumulation in hepatocytes only, the progressive forms also exist as hepatocyte injuries, apoptosis, necrosis, and inflammation cell infiltration, which eventually exacerbate liver failure, fibrosis, and hepatocellular carcinoma [3,4,5]. Although liver biopsy has been acknowledged as the gold standard for diagnosing MASLD, its limitations, such as procedural invasiveness, sampling variabilities, and other miscellaneous complications, hinder it from widespread use [6]. Therefore, it is challenging to identify MASLD patients using a convenient and accurate method to prevent disease progression.

Conventional B-mode ultrasound (US) has been accepted as an initial screening method for detecting fatty liver since it is inexpensive and widely available [1]. In US images, hepatic steatosis appears as a diffuse increase in echogenicity due to the increased parenchymal reflectivity, along with relative lower right kidney cortex echogenicity [7]. It has a high diagnostic accuracy for moderate to severe hepatic steatosis, but its sensitivity for mild steatosis is low, ranging between 53% and 76% [8,9]. However, the evidence demonstrated that even a small increase in liver fat contents to 6 ± 2%, determined by magnetic resonance spectroscopy, may indicate worsening metabolic changes [10]. Studies have pointed out that since hepatic fibrosis increases hepatic echogenicity, which is similar to echo patterns in hepatic steatosis, increased liver stiffness is associated with false-negative results in steatosis detection by US. In patients with NAFLD and mild fibrosis, US can detect steatosis with 100% sensitivity, but the sensitivity is reduced to 77.8% in those with severe fibrosis [11]. Thus, it is of critical importance to optimize the diagnostic performance of the most widely used screening method for MASLD.

The target of this research was to identify the factors influencing the accuracy of US in screening MASLD and try to optimize its performance.

## 2. Materials and Methods

### 2.1. Study Design

This study was approved by the Clinical Research Ethics Committee of our hospital (ethical number: [2020] 187) and registered in the Chinese Clinical Trial Register (ChiCTR2000034197). Each subject provided a signed written informed consent.

In this prospective study, we recruited 1489 consecutive Chinese patients who had undergone both US and magnetic resonance imaging-based proton density fat fraction (MRI-PDFF) measurements in the fatty liver clinic of our hospital between July 2020 and January 2022. During the same period, another validation cohort containing 426 consecutive inpatients who had undergone both US and liver biopsy was established from 5 tertiary care medical centers in China.

Inclusion criteria: (1) patients over 18 years of age; (2) patients who provided complete estimation of medical history and routine laboratory tests. Exclusion criteria: (1) history of alcohol abuse (average drinking of >210 g/week for men or >140 g/week for women); (2) patients who received pharmacracy that was associated with secondary fatty liver, including tamoxifen and steroid hormone; (3) patients with abnormal function of thyroid, pituitary or adrenal, and viral hepatitis (positive hepatitis B surface antigen or antibody against hepatitis C virus); (4) with evidence of cirrhosis, liver cancer or extra-hepatic tumors.

MASLD is diagnosed by evidence of steatotic liver by MRI-PDFF or liver biopsy with any one of the cardiometabolic criteria including: (1) Body mass index (BMI) ≥ 23 kg/m^2^ or waist circumference (WC) > 94 cm (male) 80 cm (female); (2) Fasting blood glucose (FBG) ≥ 5.6 mmol/L or type 2 diabetes mellites (DM) or drug treatment for type 2 DM; (3) Blood pressure ≥ 130/85 mmHg or antihypertensive drug treatment; (4) Plasma triglycerides(TG) ≥ 1.70 mmol/L or lipid-lowering treatment; (5) Plasma high-density lipoprotein (HDL) ≤ 1.0 mmol/L (male) and ≤1.3 mmol/L (female) or lipid-lowering treatment [1].

### 2.2. Clinical and Metabolic Evaluation

Anthropometric parameters, including weight, height, WC, and hip circumference, were obtained by two experienced physicians. BMI was defined as weight (kg)/height^2^ (m^2^). The metabolic syndrome was evaluated according to the Adult Treatment Panel III (ATP-III), except that the cutoff point of waist circumference of central obesity adopted the China standard: WC ≥ 90 cm for men and ≥85 cm for women, FBG ≥ 5.6 mmol/L or 2-h postprandial blood glucose ≥ 7.8 mmol/L or with medical history of DM; TG ≥ 1.7 mmol/L or usage of lipid-lowering agents, HDL < 1.0 mmol/L for man or <1.3 mmol/L for woman, and blood pressure ≥ 130/85 mmHg or usage of antihypertensive agents [12].

After the patient had fasted for 8 h, blood samples were obtained and examined for the following indexes: alanine aminotransferase (ALT), aspartate aminotransferase (AST), alkaline phosphatase (ALP), glutamate transpeptidase (GGT), uric acid; cholesterol, TG, HDL, low-density lipoprotein (LDL), FBG, and fasting insulin (FINS). The HOMA-IR was calculated as fasting blood glucose(mmol/L) × fasting blood insulin (μU/mL)]/22 [13].

Five invasive indexes were calculated with the following formulae:
Fatty liver index (FLI): (e^0.953×loge (TG)+0.139×BMI+0.718×loge (GGT)+0.053×WC−15.745^)/(1 + e^0.953×loge (TG/0.0113)+0.139×BMI+0.718×loge (GGT)+0.053×WC−15.745^) × 100 [14]Hepatic steatosis index (HSI):8 × ALT/AST ratio +BMI (+2, if DM; +2, if female) [15]Liver Fat Score (LFS): −2.89 + 1.18 × metabolic syndrome (yes = 1/no = 0) + 0.45 × Type 2 DM (yes = 2/no = 0) + 0.15 × FINS (mU/L) + 0.04 × AST(U/L) − 0.94 × AST/ALT [16]Visceral Adiposity Index (VAI) [17]Male: VAI = [WC/39.68 + (1.88 ×BMI)] × [TG/1.03] × (1.31/HDL)Female: VAI = [WC/36.58 + (1.89 × BMI)] × (TG/0.81) × (1.52/HDL)Triglycerides × Glucose index (TyG): ln [TG (mg/dL) × FBG (mg/dL)/2] [17]

### 2.3. Ultrasonography

Ultrasonography was performed within 2 weeks after serum assays by three doctors with special training. Typical ultrasonography features of fatty liver include the presence of bright liver, hepatorenal echo contrast, hepatosplenic echo contrast, vascular blurring, and attenuation [18].

### 2.4. Two-Dimensional Shear Wave Elastography (2D-SWE)

Subjects were asked to fast for at least 4 h before examination. Two-dimensional shear wave elastography was carried out using a supersonic imaging system (Aix-en-Provence, France) by two fixed sonographers with over 3 years of experience. They were blinded to the aim of the study and other clinical information. SWE was performed in dual mode using the right intercostal approach for image collection. Elastography measurements were obtained with regions of interest (ROI, approximately 4 cm × 3 cm) placed in the center of the image near the focus and at least 2 cm below the liver capsule, avoiding rib shadows and major intrahepatic vasculature or general artifacts. All acquisitions were obtained during a brief breath-hold of 8 to 10 s after inspiration to avoid motion artifacts. Five consecutive 2D-SWE images were acquired, and the mean values were used for analysis [19].

### 2.5. MRI-PDFF

MRI-PDFF was carried out with a 3.0-Tesla MRI scanner (SIEMENS 3.0 T MAGNETOM Verio). T1 volumetric interpolated breath-hold examination Dixon sequence was used to obtain fat–water separation images. The parameter settings were as follows: TE1 2.5 ms; TE2 3.7 ms; repetition time 5.47 ms; 5° flip angle; ±504.0 kHz per pixel receiver bandwidth; a slice thickness of 3.0 mm [13]. The average liver fat content (LFC) values were referred to classify steatosis grades. According to a meta-analysis, we used LFC of 5.36%, 15.36%, and 20.35% as cutoff values for mild, moderate, and severe steatosis grades [20].

### 2.6. Histological Assessment

Percutaneous liver biopsy was performed with 18 G Temno needles under the guidance of ultrasound. Liver specimens with at least 15 mm in length were acquired from the right hepatic lobe. Two seasoned pathologists with at least 10 years of experience and blinded to US results scored liver specimens based on the steatosis activity fibrosis scoring system (SAF) system. Steatosis was graded into four groups by the ratio of steatosis in hepatocytes <5%, <33.3%, 33.3–66.6%, and >66.6% for steatosis grade S0–S3, respectively. When there was discordance between the two pathologists, a third one would participate in the discussion and reach a final consensus.

### 2.7. Statistical Analysis

All statistical analyses were conducted using SPSS version 26.0. For continuous variables, we used mean ± standard deviation or median ± interquartile range, where appropriate. The chi-square test was performed for qualitative data. Moreover, normally distributed numerical variables were compared by Student’s *t*-test. Otherwise, the Mann–Whitney U-test was used. The area under the receiver operating characteristic (AUROC) was employed to assess the performance of the diagnostic method. Logistic regression analysis was performed to analyze the variables independently associated with US performance. A two-sided *p*-value of less than 0.05 was considered statistically significant.

## 3. Results

### 3.1. Patient Characteristics

A total of 2012 subjects completed the MRI-PDFF examination; ultrasound and laboratory tests were initially included, and 97 patients were subsequently excluded for the following reasons: 24 were for the positive hepatitis B surface antigen, 15 had a history of alcohol abuse, 12 displayed cirrhosis, 4 were taking drugs that could induce liver steatosis, 20 refused to undergo MRI-PDFF measurements or liver biopsy, 17 lacked biochemical tests, 2 had extra-hepatic neoplasia, and 3 had a poor-quality specimen. Finally, 1489 subjects were included in the training set, and 426 patients were recruited to the validation set (Figure 1). In the training set, 233 patients were classified as non-MASLD and 1256 as MASLD, confirmed by MRI-PDFF. A total of 145 out of the 233 non-MASLD patients were misdiagnosed with MASLD by US. Among the 1256 MASLD patients, 77 were misdiagnosed by US; 90% of them displayed mild MASLD. The AUROC of ultrasound was lower than that of the FLI in the research cohort (0.66 vs. 0.76, *p* < 0.001) (Table 1).

### 3.2. Clinical Characteristics of Subjects with Different Outcomes by Ultrasound

Compared with the non-misdiagnosed patients, the patients in the misdiagnosis group showed a higher level of uric acid (381 ± 89 vs. 346 ± 76 μmol/L, *p* = 0.002). We stratified the indexes into Q1, Q2, Q3, and Q4 according to the 25%, 50%, and 75% quantiles. As shown in Figure 2, the patients with a waist circumference in the Q3 zone were more likely to be misdiagnosed by ultrasound compared with those with a waist circumference in the Q4 zone (*p* = 0.04). Higher liver stiffness, measured with SWE, was associated with misdiagnosis of ultrasound. The BMI, thickness of subcutaneous fat, VAI, and TG were not associated with a misdiagnosis of ultrasound.

Compared with the non-missed diagnosis group, the proportion of patients with metabolic syndrome was lower in the missed diagnosis group (23% vs. 59%, *p* < 0.001). The MASLD patients missed by ultrasound had lower levels of ALT, TG, and FBG. The VAI, TG, and the number of metabolic dysfunction components were associated with missed diagnosis by ultrasound. A level of TG in Q1 or Q2 was associated with a missed diagnosis by ultrasound in all MASLD patients and patients with mild steatosis (*p*-value was 0.003, 0.02, respectively. Figure 3 and Figure 4) and patients with five components of metabolic dysfunction were the least likely to have their diagnoses missed (*p* = 0.003, Figure 3).

### 3.3. Factors Associated with Missed and Misdiagnosis with Ultrasound and Foundation of the Optimized Diagnostic Method

Uric acid, >1 × ULN, and cholesterol > 5.7 mmol/L were associated with a misdiagnosis with ultrasound (Figure 5). Furthermore, we analyzed the factors associated with MASLD and found that the indexes associated with metabolic dysfunction and high levels of ALT and uric acid were closely associated with MASLD (Table 2).

Since these indexes had a linear correlation with each other, they could not all be included in the foundation of a new algorithm. The number of metabolic syndrome components could reflect the overall metabolic situation; thus, this measure was chosen as a representative of metabolic dysfunction. Since the levels of ALT and uric acid were independent of metabolic dysfunction, they were included in the foundation of the new algorithm. Finally, a new algorithm combining these factors was proposed. New algorithm = 0.84 × number of metabolic dysfunction components + 0.79 (if ALT > 1 × ULN) + 2.06 (if ultrasound diagnosed with liver steatosis) + 0.63 (if uric acid > 1 × ULN) − 2.23. The AUROC of the new algorithm was 0.84, much higher than that of FLI and US alone.

### 3.4. Validation of the New Strategy in the Liver Biopsy Cohort

In the validation cohort, 200 patients were classified as MASLD and 226 as non-MASLD by liver biopsy. According to liver biopsy, 41 out of the 226 non-MASLD patients were misdiagnosed, and 33 MASLD subjects were not detected with US. In the logistic regression, TG > 1.7 mmol/L and impaired glucose metabolism were found to be positively associated with missed diagnosis of ultrasound. Abdominal obesity, BMI ≥ 25 kg/m^2^, hypertension, TG > 1.7 mmol/L, metabolic syndrome, and a high level of uric acid were positively associated with the misdiagnosis of US, while SWE > 7 kpa was negatively associated with a misdiagnosis (Figure 5). With the new algorithm, the AUROC of US was elevated significantly from 0.83 to 0.92 (*p* < 0.001, Figure 6).

## 4. Discussion

Using a cohort of 1489 subjects with MRI-PDFF examination, we investigated the diagnostic performance of conventional B-mode ultrasonography for MASLD patients confirmed by MRI-PDFF. Furthermore, we explored the factors influencing the diagnostic result obtained using ultrasound and proposed a new algorithm to optimize its accuracy, which was validated by a cohort of 426 biopsy-confirmed MASLD patients. As far as the authors know, this is the first study aiming at improving the efficacy of US in diagnosing MASLD with laboratory indexes. Our research revealed that the number of metabolic dysfunction components, ALT > 1 × ULN, and a high level of uric acid were associated with a missed diagnosis. By combining these factors with ultrasound, the US performance could be improved.

Although liver biopsy is the reference standard for the diagnosis and grading of hepatic steatosis, it is not feasible to use biopsy for continuous surveillance because of its high sampling variability and invasive nature. Conventional computed tomography (CT) can detect moderate to severe steatosis, but it cannot accurately diagnose mild steatosis. A study showed that the sensitivity and specificity of CT for mild steatosis (histological liver fat content < 20%) were 57% and 88%, respectively [21]. Moreover, CT is expensive and involves radiation exposure, which hinders its widespread use. As a result, CT is not recommended as a primary modality for measuring liver steatosis [22]. MRI-PDFF, defined as the ratio of the mobile proton density from triglycerides and the total mobile proton density from triglycerides and water, is the most promising method with excellent results [23]. Many studies have demonstrated that the liver fat quantified by MRI-PDFF is statistically concordant with a liver biopsy assessment. A meta-analysis including 1100 patients with chronic liver disease showed that the AUROC values of MRI-PDFF for classifying any grade of histological steatosis were more than 0.91 [24]. Furthermore, MRI scanners can identify nonalcoholic steatohepatitis patients who are at great risk of disease progression; thus, their utilization helps to guide the appropriate use of clinical interventions [25]. However, the cost of MRI is higher than that of ultrasound and CT, and MRI’s suitability is prone to be limited by patient factors such as claustrophobia, implanted devices, and discomfort. Compared with MRI, the advantages of a US-based hepatic fat system include its universal availability, real-time capability, ability to accommodate most patients (such as those with claustrophobia or morbid obesity who cannot fit inside an MRI bore), and lower cost. Thus, it is important to improve the performance of ultrasound to facilitate its clinical use in primary hospitals.

Considerable efforts have been directed toward improving US performance in diagnosing MASLD, especially quantitative US techniques. Recently, a new technical method, controlled attenuation parameter (CAP), through vibration-controlled transient elastography, has enabled the quantitative assessment of liver steatosis [26]. The CAP method, which is based on transient elastography, is the most widely studied quantitative US approach. A meta-analysis investigated 2346 patients with chronic liver disease and demonstrated that the accuracy of CAP in detecting the presence of any histological steatosis in patients with NAFLD is good, with an AUROC of 0.82 [27]. Nonetheless, CAP has a high rate of measurement failure (0–24%) [28], especially in patients with obesity and metabolic syndromes [29]. Recently, Remi Collin found that hepato-renal index using conventional US was superior to continuous CAP using FibroScan (Echosens, Paris, France) in diagnosing hepatic steatosis (AUROC 0.91 vs. 0.79) [30]. Although CAP has been extensively validated to have excellent diagnostic performance for detecting hepatic steatosis, this new technology is not as widely available as traditional ultrasonography, which makes it less appealing, especially in undeveloped regions [31]. Thus, conventional B-mode ultrasonography is still the first choice of screening tool for MASLD. However, the fact that the quantitative US techniques were from many vendors paradoxically hinders dissemination because of the challenge of harmonizing methods across vendors and platforms [32].

Uric acid is generated in the liver and is the predominant metabolite of purine. The production and discharging balance of uric acid is of critical importance to health. Extensive epidemiological studies have demonstrated that an elevated level of uric acid promotes the development of gout and other chronic metabolic diseases, such as cardiovascular disease and metabolic syndromes [33,34]. Additionally, emerging evidence has suggested that high levels of uric acid contribute to the development and progression of MASLD. A prospective cohort study, after following up with 2832 subjects for 4 years, reported that elevated uric acid is an independent predictor for NAFLD [35]. A cross-sectional survey in China, including 21,798 subjects, observed that uric acid levels had a significant association with ultrasound-based NAFLD, which was independent of other metabolic risk factors [36]. A previous study by our team revealed that the prevalence of severe steatosis diagnosed with ultrasound or MRI-PDFF increased with uric acid levels, and subjects with higher levels of uric acid had higher liver fat content measured with MRI-PDFF, which indicated that uric acid could be established as a risk factor for MASLD [37].

The strength of this research was that a large population of 1648 patients and an independent biopsy-confirmed cohort were included. In addition, we applied MRI-PDFF as a reference to assess the efficacy of ultrasound, which was accurate. The limitation of this research was that the non-MASLD patients were much less compared with the MASLD patients, which may affect the diagnostic accuracy of ultrasound.

## 5. Conclusions

In conclusion, this research evaluated and optimized the performance of ultrasonography for MASLD. Our research revealed that the number of metabolic dysfunction components and alanine aminotransferase > 1 × ULN were negatively associated with a missed diagnosis. Once ultrasound was combined with these factors, its performance largely improved.

## Figures and Tables

**Figure 1 diagnostics-13-03503-f001:**
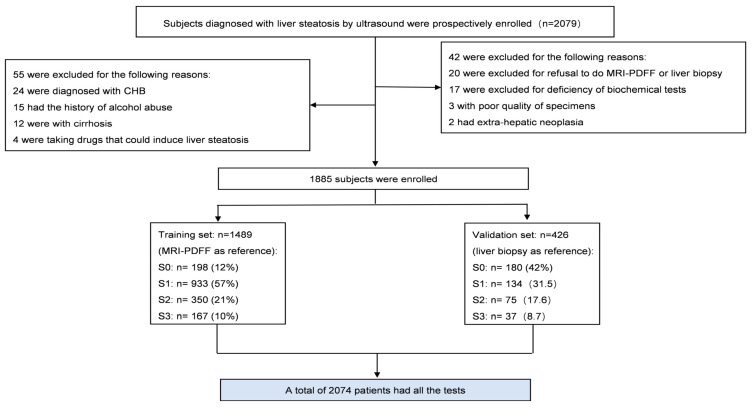
Flow chart of recruitment of patients. S0, no liver steatosis (<5.36% for MRI-PDFF or <5% for liver biopsy); S1, mild liver steatosis (5.36–15.36% for MRI-PDFF or >5–33.3% for liver biopsy); S2, moderate liver steatosis (15.36–20.35% for MRI-PDFF or >33.3-66.6% for liver biopsy); S3, severe liver steatosis (>20.35% for MRI-PDFF or >66.6% for liver biopsy). CHB, chronic hepatitis B. MRI-PDFF, magnetic resonance imaging-based proton density fat fraction.

**Figure 2 diagnostics-13-03503-f002:**
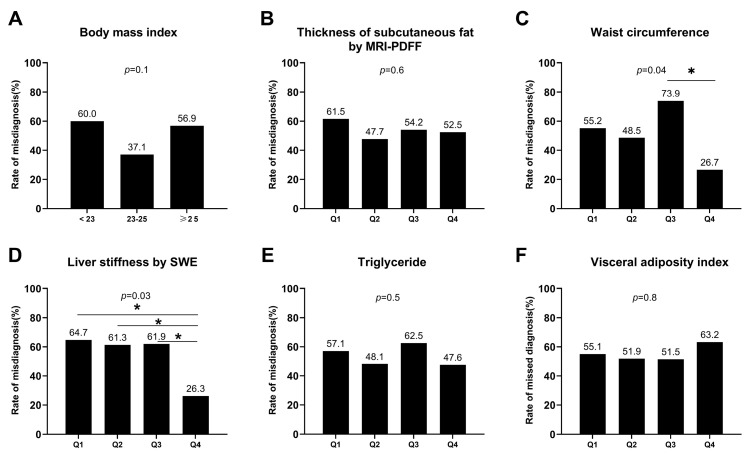
Misdiagnosis rate in different groups. (**A**) Grouped by body mass index; (**B**) Grouped by quartile of thickness of subcutaneous fat; (**C**) Grouped by quartile of waist circumference; (**D**) Grouped by quartile of liver stiffness with 2D-SWE; (**E**) Grouped by quartile of triglyceride; (**F**) Grouped by quartile of visceral adiposity index. MRI-PDFF, magnetic resonance imaging-based proton density fat fraction. SWE, 2D-shear wave elastography.*, *p* < 0.05.

**Figure 3 diagnostics-13-03503-f003:**
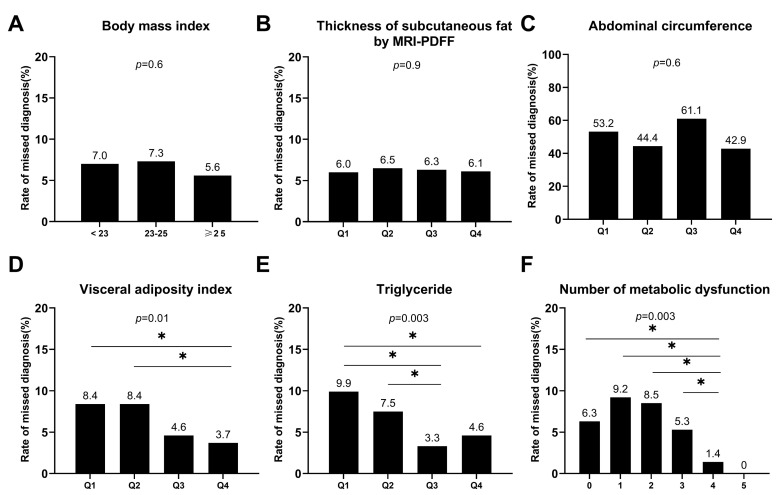
Missed diagnosis rate in different groups. (**A**) Grouped by body mass index; (**B**). Grouped by quartile of thickness of subcutaneous fat; (**C**) Grouped by quartile of abdominal circumference; (**D**) Grouped by quartile of visceral adiposity index; (**E**) Grouped by quartile of triglyceride; (**F**) Grouped by the number of metabolic dysfunctions. *, *p* < 0.05.

**Figure 4 diagnostics-13-03503-f004:**
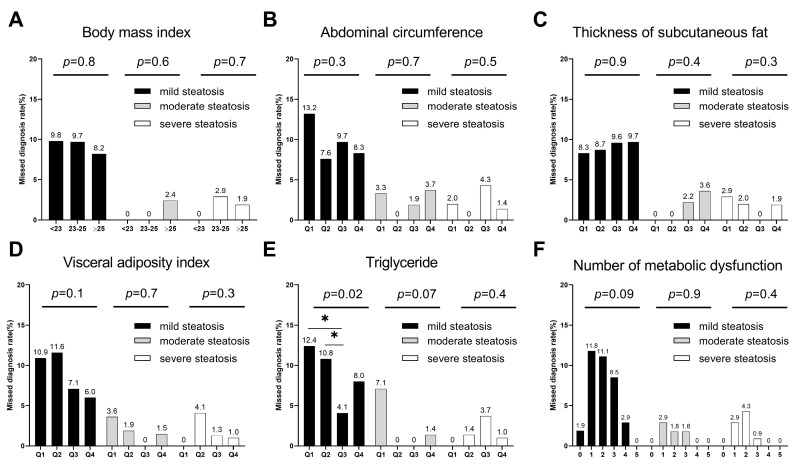
Missed diagnosis rate in different groups with different liver steatosis grades. (**A**) Grouped by body mass index; (**B**) Grouped by quartile of abdominal circumference; (**C**) Grouped by quartile of thickness of subcutaneous fat; (**D**) Grouped by quartile of visceral adiposity index; (**E**) Grouped by quartile of triglyceride; (**F**) Grouped by the number of metabolic dysfunctions. *, *p* < 0.05.

**Figure 5 diagnostics-13-03503-f005:**
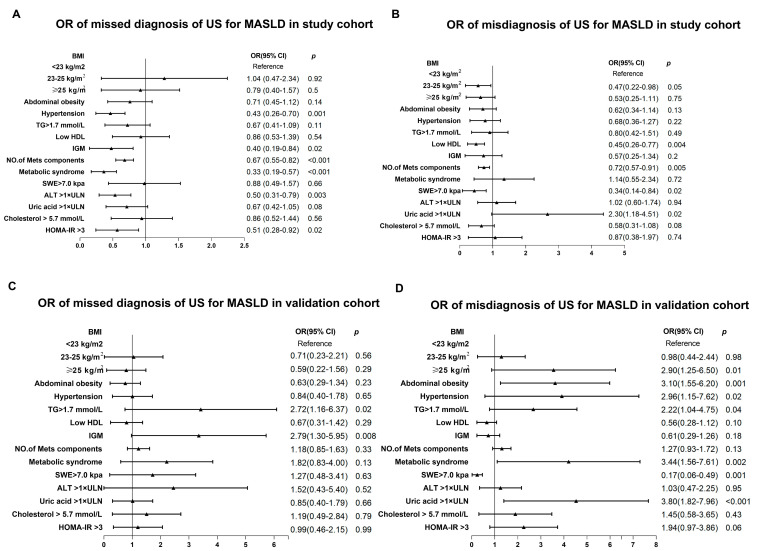
Odds ratio of factors associated with missed or misdiagnosis of ultrasound for MASLD. (**A**) Odds ratio of factors associated with missed diagnosis of ultrasound for MASLD in the study cohort. (**B**) Odds ratio of factors associated with misdiagnosis of ultrasound for MASLD in the study cohort. (**C**) Odds ratio of factors associated with missed diagnosis of ultrasound for MASLD in the validation cohort. (**D**) Odds ratio of factors associated with misdiagnosis of ultrasound for MASLD in the validation cohort. BMI, body mass index; TG, triglyceride; HDL, high-density lipoprotein; IGM, impaired glucose metabolism; SWE, liver stiffness by two-dimensional shear wave elastography; ALT, alanine aminotransferase; HOMA-IR, HOMA–insulin resistance; MASLD, metabolic dysfunction-associated fatty steatotic liver disease; CI, confidence interval; ULN, upper limit of normal.

**Figure 6 diagnostics-13-03503-f006:**
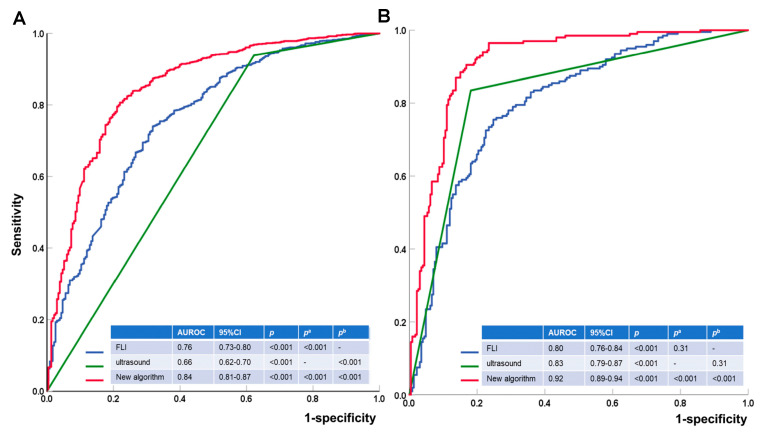
Receiver operating characteristics in different populations for MASLD. (**A**) Receiver operating characteristic for the training set. (**B**) Receiver operating characteristic for the validation set. FLI, fatty liver index. New algorithm = 0.84 × Number of metabolic dysfunctions + 0.79 (if ALT > 1 × ULN) + 2.06 (if ultrasound diagnosed with liver steatosis) + 0.63 (if uric acid > 1 × ULN) − 2.23. MASLD, metabolic dysfunction-associated fatty steatotic liver disease; CI, confidence interval. *p*^a^ for comparing with ultrasound; *p*^b^ for compared with FLI.

**Table 1 diagnostics-13-03503-t001:** Comparisons of clinical characteristics in groups of different diagnosis results by ultrasound for MASLD.

	Non-MASLD by MRI-PDFF (*n* = 233)			MASLD by MRI-PDFF (*n* = 1256)			Non-MASLD by Biopsy (*n* = 226)			MASLD by Biopsy (*n* = 200)		
	Misdiagnosis (*n* = 145)	Non-Misdiagnosis (*n* = 88)	*p*	Missed Diagnosis (*n* = 77)	Non-Missed Diagnosis (*n* = 1179)	*p*	Misdiagnosis (*n* = 41)	Non-Misdiagnosis (*n* = 185)	*p*	Missed Diagnosis (*n* = 33)	Non-Missed Diagnosis (*n* = 167)	*p*
Age, years	41 ± 12	43 ± 12	0.11	45 ± 14	41 ± 13	0.008	48 ± 15	52 ± 12	0.06	43 ± 12	42 ± 14	0.63
Male, *n* (%)	70 (0.7)	67 (0.8)	0.12	56 (0.69)	884 (0.72)	0.53	25 (73.5)	143 (77.3)	0.63	28 (0.85)	106 (0.61)	0.008
Body mass index (BMI), kg/m^2^	23.8 ± 2.8	24.6 ± 3.5	0.12	26.4 ± 4.4	27.0 ± 4.0	0.23	27.9 ± 8.3	23.5 ± 2.8	<0.001	25.6 ± 4.4	27.9 ± 5.9	0.05
BMI < 23 kg/m^2^, *n* (%)	26 (0.26)	18 (0.21)	0.22	11 (0.14)	143 (0.12)	0.64	9 (26.5)	76 (41.1)		7 (0.21)	25 (0.14)	
BMI 23–24.99 kg/m^2^, *n* (%)	14 (0.14)	20 (0.23)		15 (0.18)	188 (0.15)		7 (20.6)	69 (37.3)	<0.001	8 (0.24)	40 (0.23)	0.65
BMI ≥ 25 kg/m^2^, *n* (%)	60 (0.60)	46 (0.55)		55 (0.68)	893 (0.73)		18 (52.9)	40 (21.6)		18 (0.55)	109 (0.63)	
Diabetes mellitus (%)	4 (0.04)	5 (0.06)	0.53	4 (0.05)	97 (0.08)	0.33	2 (5.9)	16 (8.6)	0.65	5 (15.2)	24 (13.8)	0.83
Abdominal circumference, cm	86.7 ± 8.2	88.2 ± 10.2	0.32	92.6 ± 10.4	94.0 ± 9.6	0.22	97.8 ± 20.7	84.8 ± 9.0	<0.001	91.6 ± 10.0	98.8 ± 16.6	0.02
Hypertension (%)	69 (0.69)	59 (0.70)	0.93	54 (0.67)	856 (0.70)	0.52	15 (44.1)	94 (50.8)	0.55			
Systolic pressure, mmHg	125 ± 15	130 ± 16	0.02	129 ± 13	131 ± 16	0.24	127 ± 10	130 ± 13	0.08	131 ± 13	130 ± 16	0.65
Diastolic pressure, mmHg	82 ± 11	85 ± 11	0.52	85 ± 10	86 ± 12	0.46	78 ± 9	84 ± 10	0.001	87 ± 12	83 ± 11	0.04
Metabolic syndrome (%)	31 (0.31)	20 (0.24)	0.43	23 (0.28)	596 (0.49)	<0.001	13 (38.2)	22 (11.9)	0.001	22 (66.7)	91 (52.3)	0.13
Alanine aminotransferase, U/L	35.1 ± 32.0	32.9 ± 23.3	0.62	43.3 ± 36.4	58.1 ± 49.4	0.008	59.2 ± 46.5	58.1 ± 71	0.95	74.5 ± 41.6	83.4 ± 56.1	0.56
Aspartate aminotransferase, U/L	30.0 ± 21.5	30.9 ± 18.4	0.72	34.4 ± 34.0	40.1 ± 34.4	0.15	45.4 ± 29.1	64.2 ± 73.9	0.14	44.8 ± 16.3	52.3 ± 41.3	0.37
Alkaline phosphatase, U/L	75.8 ± 20.0	79.8 ± 24.7	0.23	80.7 ± 25.1	81.2 ± 31.2	0.93	104.0 ± 62.6	99.1 ± 45.6	0.63	86.0 ± 15.9	93.1 ± 46.9	0.47
Glutamate transpeptidase, U/L	47.9 ± 61.5	52.2 ± 60.7	0.63	55.5 ± 69.3	64.6 ± 77.4	0.35	84.6 ± 78.2	83.6 ± 78.6	0.93	105.0 ± 59.0	89.2 ± 93.1	0.37
Uric acid, μmol/L	381 ± 89	346 ± 76	0.002	398 ± 96	425 ± 104	0.02	378 ± 118	334 ± 82	0.005	413 ± 61	417 ± 132	0.84
Cholesterol, mmol/L	4.9 ± 1.0	5.1 ± 1.1	0.12	5.1 ± 1.0	5.2 ± 1.1	0.23	4.8 ± 1.0	4.7 ± 1.1	0.63	5.3 ± 0.9	5.1 ± 1.0	0.26
Triglyceride, mmol/L	1.3 ± 0.7	1.4 ± 0.7	0.11	1.6 ± 1.0	2.0 ± 1.3	0.02	1.3 ± 0.5	1.1 ± 0.4	0.002	2.5 ± 2.8	2.1 ± 1.5	0.23
High-density lipoprotein, mmol/L	1.2 ± 0.3	1.3 ± 0.3	0.24	1.3 ± 0.4	1.2 ± 0.4	0.14	1.4 ± 0.7	1.7 ± 1.8	0.34	1.1 ± 0.2	1.1 ± 0.5	0.54
Low-density lipoprotein, mmol/L	3.1 ± 0.8	3.2 ± 0.8	0.44	3.2 ± 0.7	3.2 ± 0.8	0.26	2.8 ± 0.8	2.7 ± 0.9	0.66	3.4 ± 0.8	3.2 ± 0.7	0.15
Fasting blood glucose, mmol/L	4.8 ± 0.6	5.2 ± 1.9	0.04	4.9 ± 0.8	5.2 ± 1.1	0.02	5.4 ± 1.0	5.5 ± 1.4	0.75	5.8 ± 1.0	5.5 ± 1.8	0.46
Fasting insulin, μU/ml	8.9 ± 5.0	8.5 ± 4.2	0.55	10.5 ± 6.0	12.1 ± 7.1	0.06	14.4 ± 9.0	10.5 ± 8.4	0.01	15.3 ± 5.8	16.7 ± 10.9	0.55
HOMA–insulin resistance	1.9 ± 1.1	2.0 ± 1.4	0.72	2.3 ± 1.5	2.8 ± 2.0	0.03	3.6 ± 2.7	2.6 ± 2.4	0.02	4.1 ± 1.9	4.2 ± 3.1	0.86
Liver fat content of MRI-PDFF Mild steatosis (%)	6.3 ± 4.9 -	3.9 ± 1.1 -	<0.05 -	8.9 ± 4.7 73 (0.90)	15.1 ± 8.1 737 (0.60)	<0.001	-	-	-	-	-	-
Moderate steatosis, *n* (%)	-	-	-	3 (0.04)	190 (0.16)	<0.001						
Severe steatosis, *n* (%)	-	-	-	5 (0.06)	297 (0.24)							
Thickness of subcutaneous fat, cm	22.0 ± 7.0	23.5 ± 7.6	0.43	24.6 ± 8.6	23.9 ± 8.3	0.56						
Liver stiffness with SWE, kpa,	6.0 ± 3.1	6.4 ± 2.6	0.42	6.5 ± 2.5	6.5 ± 2.9	0.94	13.6 ± 11.1	16.2 ± 8.2	0.24	6.9 ± 2.5	7.0 ± 3.6	0.93
Controlled attenuation parameter	260 ± 44	258 ± 43	0.82	262 ± 40	290 ± 49	0.002	-	-	-	-	-	-
Liver stiffness measurement	5.8 ± 2.6	6.8 ± 4.1	0.24	6.6 ± 2.7	8.0 ± 13.8	0.64	-	-	-	-	-	-
Fatty liver index	31.2 ± 21.4	38.6 ± 24.0	0.25	47.2 ± 26.2	56.5 ± 23.7	0.001	56.2 ± 32.0	35.2 ± 22.2	<0.001	65.1 ± 22.8	68.0 ± 24.1	0.54
Hepatic steatosis index	34.4 ± 5.0	34.4 ± 5.0	0.92	37.6 ± 6.3	38.9 ± 6.7	0.15	40.1 ± 12.3	32.4 ± 4.7	<0.001	39.3 ± 7.0	42.7 ± 10.1	0.07
Liver fat score	−1.1 ± 1.5	−1.1 ± 1.4	0.94	−0.44 ± 1.8	0.38 ± 2.1	0.001	0.6 ± 2.3	0.2 ± 3.3	0.54	1.4 ± 1.8	1.7 ± 2.5	0.67
Visceral adiposity index	1.6 ± 1.2	1.8 ± 1.6	0.46	1.9 ± 1.3	2.6 ± 2.4	0.01	1.7 ± 1.0	1.2 ± 0.9	0.002	3.8 ± 6.3	3.2 ± 2.6	0.37
Triglycerides × glucose index	8.4 ± 0.5	8.6 ± 0.5	0.03	8.5 ± 0.5	8.8 ± 0.6	<0.001	8.5 ± 0.4	8.4 ± 0.4	0.01	9.2 ± 0.6	8.9 ± 0.6	0.05

Values shown are *n* (%) or means ± standard deviation. *p*-values were for the Student’s *t*-test or chi-square testing between groups. MASLD, metabolic dysfunction-associated fatty steatotic liver disease; MRI-PDFF, magnetic resonance imaging-based proton density fat fraction; HOMA–insulin resistance, homeostasis model assessment of insulin resistance; SWE, shear wave elastography.

**Table 2 diagnostics-13-03503-t002:** Predictors of MASLD.

Variables	Study Cohort						Validation Cohort				
	Univariate			Multivariate			Univariate			Multivariate	
	OR (95% CI)	*p*		OR (95% CI)	*p*		OR (95% CI)	*p*		OR (95% CI)	*p*
Body mass index (BMI)		<0.001						<0.001			
BMI < 23 kg/m^2^	reference						reference				
BMI 23–24.99 kg/m^2^	1.67 (0.96–2.92)	0.07					1.67 (0.96–2.92)	0.07			
BMI ≥ 25 kg/m^2^	6.64 (3.94–11.2)	<0.001					6.64 (3.94–11.2)	<0.001			
Abdominal obesity	5.10 (3.37–7.72)	<0.001					5.10 (3.37–7.72)	<0.001			
Hypertension	3.61 (2.60–5.03)	<0.001					13.34 (7.92–22.50)	<0.001			
Triglyceride > 1.7 mmol/L	2.41 (1.72–3.35)	<0.001					3.94 (2.57–6.05)	<0.001			
Low–High-density lipoprotein	2.16 (1.62–2.87)	<0.001					1.42 (0.97–2.09)	0.07			
Impaired glucose metabolism	2.41 (1.55–3.75)	<0.001					0.98 (0.66–1.45)	0.92			
Number of Mets components	2.52 (2.16–2.93)	<0.001		2.32 (1.96–2.76)	<0.001		2.75 (2.22–3.41)	<0.001		2.43 (1.87–3.16)	<0.001
Metabolic syndrome	4.28 (2.96–6.18)	<0.001					7.54 (4.77–11.90)	0.002			
Alanine aminotransferase > 1 × ULN	3.66 (2.74–4.88)	<0.001		2.20 (1.58–3.08)	<0.001		2.54 (1.49–4.30)	0.001		2.25 (1.00–5.05)	<0.001
Uric acid > 1 × ULN	3.47 (2.53–4.77)	<0.001		1.87 (1.31–2.69)	<0.001		5.40 (3.52–8.30)	<0.001		2.68 (1.49–4.83)	0.001
Cholesterol > 5.7 mmol/L	1.43 (1.02–1.99)	0.04					1.84 (1.11–3.07)	0.02			
HOMA–insulin resistance > 3.0	3.15 (2.07–4.79)	<0.001					3.59 (2.40–5.35)	<0.001			

Hypertension, blood pressure > 130/85 mmHg or using antihypertensive drugs; Abdominal obesity: >90 cm for males or >85 cm for females; Low–high-density lipoprotein: <1.0 mmol/L for males or <1.3 mmol/L for females; Impaired glucose metabolism, fasting blood glucose ≥ 5.6 mmol/L or 2-h post-meal blood glucose ≥ 7.8 mmol/L or with type 2 diabetes mellitus; Number of Mets components, number of the following: abdominal obesity, hypertension; triglyceride > 1.7 mmol/L, low–High-density lipoprotein, and impaired glucose metabolism. MASLD, metabolic dysfunction-associated fatty steatotic liver disease; Mets, Metabolic syndrome; CI, confidence interval; HOMA–insulin resistance, homeostasis model assessment of insulin resistance; ULN, upper limit of normal.

## Data Availability

The datasets used during the current study are available from the corresponding author upon reasonable request.

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
