# Peer review of "An Optimized Strategy Based on Conventional Ultrasound for Diagnosing Metabolic Dysfunction-Associated Steatotic Liver Disease"

_diagnostics, 2023, doi:10.3390/diagnostics13233503_

Round 1
Reviewer 1 Report
Comments and Suggestions for Authors
In this paper the authors explore the role of combining conventional abdominal US with different clinical and biochemical parameters routinely analysed in patients with metabolic syndrome with the aim of better identify patients with Metabolic dysfunction associated steatotic liver disease (MASLD). Given the increasing prevalence of diabetes, obesity and metabolic syndrome worldwide and the impact that insulin resistance and metabolic disturbances have on hepatic structure and function it is of extremely importance to provide new tools for the proper diagnostic assessment of these patients. The research by Feng X et al perfectly focus this topic and suggest a new and corroborated strategy that can be easily introduced in clinical practice. The analysis is well conducted, and conclusion are supported from reported analyses. The use of a validation cohort appears correct for the end-point of the study. In my opinion the topic is original and relevant for the field. For my part, no further improvements are needed.
Author Response
Reviewer 1
In this paper the authors explore the role of combining conventional abdominal US with different clinical and biochemical parameters routinely analysed in patients with metabolic syndrome with the aim of better identify patients with Metabolic dysfunction associated steatotic liver disease (MASLD). Given the increasing prevalence of diabetes, obesity and metabolic syndrome worldwide and the impact that insulin resistance and metabolic disturbances have on hepatic structure and function it is of extremely importance to provide new tools for the proper diagnostic assessment of these patients. The research by Feng X et al perfectly focus this topic and suggest a new and corroborated strategy that can be easily introduced in clinical practice. The analysis is well conducted, and conclusion are supported from reported analyses. The use of a validation cohort appears correct for the end-point of the study. In my opinion the topic is original and relevant for the field. For my part, no further improvements are needed.
Reply: Thank you very much for your positive feedback.
Reviewer 2 Report
Comments and Suggestions for Authors
Thank you to select me as a reviewer. The authors said that the authors evaluated and optimized the performance of ultrasound for MASLD. Our research revealed that the number of metabolism dysfunction components and alanine aminotransferase >1ⅹULN were negatively associated with missed diagnosis. After combining these factors with ultrasound, the performance of ultrasound largely improved. This paper will be useful of research in MASLD.
Author Response
Reviewer 2
Thank you to select me as a reviewer. The authors said that the authors evaluated and optimized the performance of ultrasound for MASLD. Our research revealed that the number of metabolism dysfunction components and alanine aminotransferase >1ⅹULN were negatively associated with missed diagnosis. After combining these factors with ultrasound, the performance of ultrasound largely improved. This paper will be useful of research in MASLD.
Reply: We are most grateful for the time you have spent on our manuscript, and we are thankful for your positive feedback.
Reviewer 3 Report
Comments and Suggestions for Authors
Feng et al. proposed an interesting article concerning steatosis diagnosis. The work is hard, with a lot of patients and two cohorts based on different "glod-standard".
The results are clearly shown, and interpretation is OK.
Just two remarks:
- In limits, it is imprtant to underline that patients are essentially chinese. It is difficultto copare with non-Asian populations;
- References can be improved. Cf. this recent article: https://www.ncbi.nlm.nih.gov/pmc/articles/PMC10303516/pdf/WJG-29-3548.pdf
Author Response
Reviewer 3
Feng et al. proposed an interesting article concerning steatosis diagnosis. The work is hard, with a lot of patients and two cohorts based on different "glod-standard".
The results are clearly shown, and interpretation is OK.
Just two remarks:
- In limits, it is imprtant to underline that patients are essentially chinese. It is difficultto copare with non-Asian populations;
Reply: We appreciate your kind reminding very much。And we have added the information that patients are Chinese. (line 40, line 86)
-References can be improved. Cf. this recent article: https://www.ncbi.nlm.nih.gov/pmc/articles/PMC10303516/pdf/WJG-29-3548.pdf
Reply: Thank you for the important point. In this article, Remi Collin et al evaluated new ultrasonographic tools to detect hepatic steatosis using MRI-PDFF as gold standard. And they found that hepato-renal index (HRI) calculation performed best among all ultrasonographic tools, including new-generation systems such as continuous controlled attenuation parameter and sound speed examination. The findings of this research added great value to the evaluation of ultrasound in hepatic steatosis, which is also the main topic of our research. Thus, we added the article in our references. (line 343-345,464-466)
Reviewer 4 Report
Comments and Suggestions for Authors
I liked your research and should say I am impressed with this. Just some minor suggestions and comments:
1) In line 71-73: Please clarify this statement. “Besides, the concurrent liver fibrosis may influence the detection of steatosis for they share similar echo patterns [11].”2) In line 129: It should be “presence of” instead of only “presence”3) In line 211: Did you mean ‘misdiagnosis’?4) In line 283-284: “Finally, a new algorithm combining these factors were proposed.” Should be “m. Finally, a new algorithm combining these factors was proposed.”
Comments on the Quality of English Language
Minor edits would make it a really strong article to publish.
Author Response
Reviewer 4
I liked your research and should say I am impressed with this. Just some minor suggestions and comments:
1) In line 71-73: Please clarify this statement. “Besides, the concurrent liver fibrosis may influence the detection of steatosis for they share similar echo patterns [11].”
Reply: We are very sorry for the vague expression. Researches have pointed out that since hepatic fibrosis increases hepatic echogenicity, which is similar with echo patterns in hepatic steatosis, increased liver stiffness is associated with false-negative results in steatosis detection by US [1]. In patients with NAFLD and mild fibrosis, US can detect steatosis with 100% sensitivity, but the sensitivity reduced to 77.8% in those with severe fibrosis [2]. We have made the statement clearer. (line 72-77)
[1] Dietrich CF, Shi L, Löwe A, Dong Y, Potthoff A, Sparchez Z, et al. Conventional ultrasound for diagnosis of hepatic steatosis is better than believed. Z Gastroenterol. 2022 Aug;60(8):1235-1248. English. doi: 10.1055/a-1491-1771.
[2] Lupşor-Platon M, Stefănescu H, Mureșan D, Florea M, Szász ME, Maniu A, et al. Noninvasive assessment of liver steatosis using ultrasound methods. Med Ultrason. 2014 Sep;16(3):236-45. doi: 10.11152/mu.2013.2066.163.1mlp.
2) In line 129: It should be “presence of” instead of only “presence”
Reply: Thank you for your kind reminding. We have corrected the expression. (line 133-134)
3) In line 211: Did you mean ‘misdiagnosis’?
Reply: We are sorry for our mistake and appreciate your kind suggestion very much. We have corrected the wrong word. (line 203)
4) In line 283-284: “Finally, a new algorithm combining these factors were proposed.” Should be “m. Finally, a new algorithm combining these factors was proposed.”
Reply: We appreciate your kind suggestion very much. We have corrected the wrong word. (line 279)
- Comments on the Quality of English Language: Minor edits would make it a really strong article to publish.
Reply: Thank you for your kind reminding. We have had English editing for our article at https://www.mdpi.com/authors/english. (ID: english-73679). And we have made necessary corrections according to the edition suggestion.